# Irisin and Autophagy: First Update

**DOI:** 10.3390/ijms21207587

**Published:** 2020-10-14

**Authors:** Mirko Pesce, Patrizia Ballerini, Teresa Paolucci, Iris Puca, Mohammad Hosein Farzaei, Antonia Patruno

**Affiliations:** 1Department of Medicine and Aging Sciences, University G. d’Annunzio, 66100 Chieti, Italy; mirko.pesce@unich.it (M.P.); antonia.patruno@unich.it (A.P.); 2Department of Neurosciences, Imaging and Clinical Sciences, University G. d’Annunzio, 66100 Chieti, Italy; 3Department of Oral, Medical and Biotechnological Sciences, University G. d’Annunzio, 66100 Chieti, Italy; teresa.paolucci@uniroma1.it; 4Sport Academy SSD, 65010 Pescara, Italy; pucairis@gmail.com; 5Pharmaceutical Sciences Research Center, Health Institute, Kermanshah University of Medical Sciences, 67146 Kermanshah, Iran; mh.farzaei@gmail.com

**Keywords:** irisin, FNDC5, myokine, physical activity, autophagy

## Abstract

Aging and sedentary life style are considered independent risk factors for many disorders. Under these conditions, accumulation of dysfunctional and damaged cellular proteins and organelles occurs, resulting in a cellular degeneration and cell death. Autophagy is a conserved recycling pathway responsible for the degradation, then turnover of cellular proteins and organelles. This process is a part of the molecular underpinnings by which exercise promotes healthy aging and mitigate age-related pathologies. Irisin is a myokine released during physical activity and acts as a link between muscles and other tissues and organs. Its main beneficial function is the change of subcutaneous and visceral adipose tissue into brown adipose tissue, with a consequential increase in thermogenesis. Irisin modulates metabolic processes, acting on glucose homeostasis, reduces systemic inflammation, maintains the balance between resorption and bone formation, and regulates the functioning of the nervous system. Recently, some of its pleiotropic and favorable properties have been attributed to autophagy induction, posing irisin as an important regulator of autophagy by exercise. This review article proposes to bring together for the first time the “state of the art” knowledge regarding the effects of irisin and autophagy. Furthermore, treatments on relation between exercise/myokines and autophagy have been also achieved.

## 1. Introduction

The myofibers are cells multinucleated that composed the skeletal muscle (SkM). These cells are responsible of the force generation by sarcomeres’ contraction and release several soluble factors (e.g., cytokines) exerting paracrine and endocrine functions. These molecules are named “myokines” [1,2].

Myokines regulate a variety of metabolic processes in various tissues and organs, such as liver, bones, brain, or fat tissue, through several signaling pathways. The extent of the myokines release response after muscular contraction varies with the intensity, mode, and volume of exercise an individual performs. Some myokines may be anabolic and have direct growth-promoting effects. Still others generate signals that may mediate some of the health benefits of physical exercise [3,4,5]. Since their discovery in 2000 [6], myokines have continued to generate increasing interest.

In 2012, a new myokine expressed through the transcription factor peroxisome proliferator-activated receptor gamma co-activator-1α (PGC-1α) activation by exercise-induced effects, was discovered. The newly identified molecule has been called “irisin,” derived from the name of the Greek goddess Iris. It is released as a hormone-like myokine as protein cleaved from transmembrane protein precursor fibronectin type III domain-containing protein 5 (FNDC5) encoded by the FNDC5 gene. This hormone induces changes in the subcutaneous adipose tissue stimulating browning. In general, irisin causes a significant increase in total body energy expenditure and resistance to obesity-linked insulin-resistance (IR). Irisin function summarizes some of the most important benefits of exercise and muscle activity. It was proposed to act as a link between the muscles and other tissues of the body [7].

Autophagy is a catabolic process that has an essential role in cellular homeostasis, facilitating lysosomal degradation and recycling of intracellular misfolded proteins and injured organelles. It is involved in the maintenance of various physiological responses and plays dual roles in inducing cytoprotection and cell death [8]. Regulation of autophagy formation is tightly controlled to maintain nutrient and energy homeostasis. The process is well conserved in organisms, and in response to nutritional deprivation or intracellular pathogens, it acts as a protective mechanism that allows cells to survive under stressful conditions. The process begins with an expanding membrane structure which leads to formation of a double-membrane vesicle, called autophagosome. Autophagosomes are subsequently fused with lysosomes, in which the resulting molecules are degraded and used in the recycling process. Abnormal regulation of autophagy may be an attribute to various pathologic conditions [9] and has been widely implicated in cancer, metabolic disorders, and neurodegenerative diseases, as well as muscle dystrophy. It also plays an important role in aging [10].

Regular physical activity is considered as a nondrug therapy for the treatment of several diseases, such as metabolic diseases [11]. Its beneficial effects on adults’ health has been related also to the induction of autophagy. Recent findings indeed suggested that exercise can induce autophagy in several tissues (e.g., adipose tissue and SkM) and organs (e.g., pancreas, liver, heart, and brain), supporting the evidence that exercise-mediated induction of autophagy could occur at systemic level [12,13,14].

Since then, irisin has been the subject of extensive research, which enabled the gaining of insight into its pleiotropic properties. The role of irisin on autophagy has been studied only recently, particularly during the last 3 years. However, these recent studies have not yet been reviewed. Given the emerging importance of irisin and autophagy in mediating the beneficial effects of exercise, in this review, we perform a first update on studies focused on the relation within FNDC5/irisin system and autophagy process. As irisin is a myokine induced after physical exercise, relation between exercise and autophagy as well as myokines and autophagy have been also reviewed.

## 2. Molecular Mechanism of Autophagy

Autophagy is recognized as a vital process in the cells that plays important role in biological evolution, immune system, and cell death and is involved in fatal disorders such as nervous system defects, autoimmune diseases, and a variety of cancers. Autophagy has a dual function, on one hand, it increases the duration and survival of the cells and on the other hand, in the advanced stages, it causes cell death [15].

In detail, autophagy is a degradative process including lysosomes, mainly involved in degradation of damaged cells organelles to recycle the basic cellular components, in that way providing the energy for cell recovery. The autophagy impairment is characterized by the insufficiency to remove damaged organelles or debris. The process activation was originally identified in response to starvation in eukaryotic systems. The process is evolutionary, highly conserved, and play a housekeeping role in maintaining the integrity of intracellular proteins and organelles in many cell types. Recent studies have shown that the key role of autophagy is to keep cells alive under stressful conditions, including endoplasmic reticulum stress, infection, and hypoxia [16]. Activation of autophagy pathway in SkM during exercise training as a protective mechanism can also be effective in adaptation and improvement of physical performance [17].

Exercise is indeed a new defined stimulus that can induce autophagy in cells, and its relationship with autophagy is treated with more details in Section 5. Additionally, studies have shown that autophagy increases in a variety of liver injuries, e.g., it can increase cell survival during liver ischemia reperfusion, acute liver injury and liver surgery, improvement of hepatocellular carcinoma by its antitumor activity and improvement of fatty liver disease by lipid degradation to fatty acid and modulation of insulin signaling pathway [18]. In another study that evaluates the role of autophagy in renal physiology states that autophagy is involved in improvement of kidney defects such as diabetic nephropathy, tubular injuries, kidney development, and kidney cancers [19].

There are three defined types of autophagy: macroautophagy, microautophagy, and chaperone-mediated autophagy, all of which promote proteolytic degradation of cytosolic components at the lysosome [16]. Here, macroautophagy is referred to as autophagy.

Diverse molecular signals are involved in the regulation of autophagy. At transcriptional level, regulation includes the transcription factors nuclear factor kappa B (NFkB), hypoxia-inducible factor-1 (HIF-1), and Forkhead Box O (FOXOs). The process is regulated by a highly conserved family of proteins referred to as autophagy-related genes (Atgs) and is mainly divided into different and coordinated stages: initiation, vesicle nucleation, vesicle elongation, vesicle retraction, and finally, fusion and degradation [20]. In the initiation stage of autophagy, there is the activation of the Unc-51-like autophagy activating kinase 1 (ULK1) complex and two protein factors, including focal adhesion kinase family interacting protein of 200 kDa (FIP200) and Atg13 responsible for maintaining the stability of ULK1 protein, and its correct localization to autophagy precursors [21]. When nutrients are deficient, AMP-activated protein kinase (AMPK, a sensor for detecting nutrition and energy efficiency in tissues) can directly phosphorylate the Raptor protein at Ser722 and Ser792 sites, inhibit mammalian target of rapamycin (mTOR) signaling, and deactivate the phosphorylation of ULK1, thereby activating autophagy [22]. The activity of ULK1 is enhanced by direct phosphorylation of AMPK. As a consequence, the formation of autophagic bilayer membrane is hasten following the formation of the complex including ULK1-Atg13-FIP200 [23]. On the other hand, under nutrient-rich conditions, mammalian target of rapamycin complex 1 (mTORC1) phosphorylates ULK1 at Ser757, to inhibit autophagy initiation [16]. During the second stage, the complex Beclin1-class III phosphoinositide 3-kinase (PI3K) plays a significant role in the induction of autophagy acting on the recruitment of lipids and proteins from the cytoplasm for the synthesis of autophagosome membranes [24]. To follow, the extension of autophagosome membrane is mainly controlled by Atg7 and Atg10. Atg7 is responsible (and marker) of autophagic vasculogenesis and elongation regulating the formation of autophagic bubbles. Under this stage, Atg7 can catalyze the conversion of critical protein LC3I to LC3II that is considered the “hallmark” biomarker for evaluating the activation of autophagy. Through vesicle retraction stage, the mature autophagosome is finally formed recruiting Atg12-Atg5-Atg16L complex. Ultimately, the autophagolysosomes were formed by the fusion of lysosomes and autophagosomes, also through the involvement of the lysosomal-associated membrane protein 2 (LAMP2). The autophagolysosomes are responsible of degradation of damaged macromolecules and dysfunctional organelles in cells for the recycling; the contents are broken down into constituent macromolecular precursors that can be reused as raw “bio”-material or, alternatively, metabolized (fusion and degradation stage) [25]. A summary of the molecular events involved in the autophagy process is illustrated in Figure 1.

Nevertheless, some studies approve the inhibition of binding process between autophagosome and lysosome to result in the decrease in lysosomal degradation rate of autophagosomes and upstream the proportional expression of LC3II and LC3II/LC3I ratio. This indicates that the augmented expression of LC3II does not completely represent the occurrence of autophagic degradation process [26].

The autophagic adaptor p62/SQSTM1 permits to recycle nutrients and ATP by the recognition, division, and transfer of p62 or the degradation of its substrate. In this stage, p62 is negatively correlated to the autophagic flux downstream the lysosome activity of autophagolysosomes [27]. In summary, it is broadly accepted that the combination of expression level of p62 and the conversion rate of LC3 could be considered the best biomarkers for evaluating the functional status of autophagy.

## 3. Myokines

The SkM has the function of maintaining body temperature, generating force, and maintaining posture interacting with bone and facilitating various movements. This tissue possesses the ability to adapt to various physiologic conditions such as chronic hypokinesia or exercise [28]. Nevertheless, during the last years, a great deal of data highlighted that the SkM tissue is the main protein reserve of the body [29], a regulator of health and survival, acts on metabolic balance, and finally, determines the quality of life [30]. The plasticity of SkM confers the tissue the property to constantly adapt to various settings such as hypokinesia, leading to maintenance of muscular atrophy (reduction in muscle mass), or mechanical stimulation, leading to the condition of maintenance of muscular hypertrophy (increase in muscle mass). SkM has also an elevated capacity to modify its phenotype in relation to the mechanical load applied to it [31]. Pedersen et al. (2007) first identified that SkM is additionally an organ that release a great amount of molecules such as cytokines and other peptides, which have been given the name of myokines [2]. After the discovery that muscle contraction produces these molecules, in 2013, Pedersen defined a new paradigm, i.e., SkM is a secretory organ, synthesizing and secreting myokines in response to muscle contraction. The molecules secreted can influence the metabolism and function of SkM, in addition to other tissues and organs [28,32].

The interest on the molecules released by SkM became relevant during the second half of the last century. Physical exercise was proposed as a modulator of glucose metabolism, however, with no definition about the mechanisms involved in this process [33]. The terms “exercise factor,” “work factor,” or “work stimulus” were used to define the capacity of the SkM tissue to release myokines at systemic level during or after muscle contraction. This concept was founded on the fact that the engagement of metabolic and physiological responses in other organs by muscle contractions are not mediated by the nervous system [34]. It was supported by previous study with electrical stimulation in paralyzed muscles, in which Kjaer and coworkers verified that the ability of muscle contractions in regulating other tissues occurs through an independent pathway of activation of the nervous system [35]. Therefore, a muscle memory could hypothesize independently, but in any case, related to the central nervous system (CNS). Muscle memory could organize and structure itself based on the secretion, role, and activation of the muscle myokines, which open up interesting perspectives [36]. From that moment, myokines began to be understood as a protective factor against disease, aging and the adverse effects of physical inactivity.

The level of the myokines’ plethora is promoted by both exercise at high-intensity and hypokinesia, regulating numerous physiological adaptations. Several studies have demonstrated that myokines can regulate physiological and metabolic pathways in other tissues. To date, the myokines that have been described are included of cytokines, peptides, growth factors, and small organic acids. These molecules exert an effect on muscle function itself as well as on overall metabolism and comprise myostatin (MSTN), the small organic acid β-aminoisobutyric acid (BAIBA), meteorin-like (METRNL), myonectin, decorin, various interleukins (IL-4, IL-6, IL-7, IL-8, and IL-15), irisin, fibroblast growth factor-21 (FGF-21), brain-derived neurotrophic factor (BDNF), insulin-like growth factor-1 (IGF-1), leukemia inhibitory factor (LIF), and follistatin like protein-1 (FSTL-1) [28,36,37]. Their diverse effects can be positive or negative and provide the underpinning of the crosstalk between SkM and other organs, such as adipose tissue, brain, bone, liver, pancreas, kidney, and immune cells. The SkM is the most abundant tissue in healthy individuals, constituting approximately 40% of body mass in young adults, as a consequence, any endocrine activity would be expected to have a strong effect on physiology of the organism [33]. In general, muscle atrophy is related to aberration of metabolism associated with a variety of clinical conditions, including diabetes, obesity, renal failure, chronic obstructive pulmonary disease, cancer, and muscle hypokinesia [2,28,38]. For instance, individuals are highly predisposed to obesity, IR, and diabetes mellitus that are characterized by reduced basal metabolic rate and low physical activity [1,39].

GWAS and transcriptomics analyses suggested that more than 250 putative myokines exist, 100 of which have been also identified by proteomics approaches. Recent reviews have treated the important biological effects, on specific cells and tissues, of the most studied myokines (e.g., IL-6 and irisin), highlighting their biomedical relevance [37,40,41]. Several evidences supported the idea that autophagy acts as trait d’union in this process. In Section 6, the potential role of relevant myokines (IL-4, IL-6, IL-7, IL-15, decorin, LIF, metrnl, myonectin, and myostatin) on autophagy regulation has been discussed.

## 4. Irisin Structure and Function

In 2012, Boström et al. discovered a new molecule named “irisin” secreted by myocytes, which was proposed to act as a connection between the muscles and other tissues of the body, and in particular, was able to activate thermogenesis and induce changes in adipose tissue [7]. Since then, irisin has been the subject matter of extensive research, which allowed the gaining of insight into its pleiotropic properties.

The precursor of irisin is FNDC5 (FRCP2/PeP), a cell membrane protein [7]. This protein contains an N-terminal signal sequence of 29 amino acid (aa), a fibronectin type III domain with 94 aa, an unidentified region consisting of 28 aa, a transmembrane domain having 19 aa, and a C-terminal part with 39 aa (Figure 2), for a total of 209 aa residues. The extracellular N-terminal portion is proteolytically cleaved to produce irisin that is released into the circulation, whereas the C-terminal fragment of FNDC5 is located in the cytoplasm [42]. Irisin is composed of 112 aa. Its structure has been revealed by X-ray crystallography and biochemical studies: the protein occurs in the form of homodimers, where a β-sheet is created between the units. In addition, the interactions between the side chains of adjacent subunits occurs between Arg75 and Glu79, which in turn protect the dimer ends and Trp90/Trp90, stabilizing the structure [43].

During the post-translational process of N-glycosylation, a variable number and structure of oligosaccharides (glycans) were attached to the FNDC5 protein, resulting in a mass range from 20 to 32 kDa [7]. In general, the glycosylation process is fundamental for the elicitation of the biological functions of the proteins. FNDC5 contains the sequence Asn-X-Ser/Thr in which oligosaccharides were attached to the asparagine residue by a N-acetylglucosamine residue (GlcNAc). The X indicates any amino acid apart from proline. A high-mannose/oligomannose type, complex type, and hybrid type are the main three groups that can be linked to Asn by N-glycosidic bonds [44]. The sequence of FNDC5 contains three potential N-glycosylation sites and two of them, Asn36 and Asn81, are occupied by N-glycans (Figure 2). However, the structure and the glycosylation process are still poorly characterized. The presence of glycans act significantly on the stability of the molecule. The glycosylated molecule is significantly more resistant against protein synthesis inhibitors compared to the de-N-glycosylated FNDC5. The half-life of the protein is significantly reduced, from 12 to 7 h, when one of the N-glycosylation sites is removed. In addition, the de-N-glycosylated protein is not incorporated in the plasmatic membrane, causing significant reduction in irisin release at systemic level. Irisin is also characterized by the presence of two N-glycosylation sites (Asn7 and Asn52), whereas deglycosylation lowers the molecular weight of irisin to 12 or 15 kDa. In alternative, the addition of one or two sugar chains increases its mass to 22 or 25 kDa, respectively. These modifications do not affect the formation of irisin dimers but are important to the primary function of irisin on the adipocytes [44,45].

So far, a specific receptor for irisin has not been identified. Recent studies have showed that irisin could exerts its action via binding to integrins of αv family in some tissues [46]. Integrins are transmembrane receptors ubiquitously expressed that permit to cell interactions with extracellular matrix ligands, intercellular interactions, and nevertheless recognize soluble ligands. In summary, these proteins are strictly involved in the adhesion, migration, and aggregation of cells. Twenty-four different integrin heterodimers exist, due to the interactions between 18 α-subunits and eight β-subunits [47]. In 2018, Kim and coworkers showed the binding of irisin with high affinity to αVβ5 integrin present in the fat cells and osteocytes. The use of the integrin inhibitor RGD peptide (which binds to αvβ5) suppressed any signaling response triggered by irisin. Irisin also binds other integrins including α1β1. When the osteocytes were treated with irisin, the level of phosphorylation of focal adhesion kinase (FAK) increased significantly, whereas FAK is the main protein involved in the integrin signaling. The authors suggest that the main receptors for irisin could be the heterodimers belonging to the αv integrin family in all the tissues [46].

Irisin is secreted primarily by SkM as well as subcutaneous and visceral adipose tissues. However, as showed by immunohistochemical studies, smaller amounts of irisin are also produced by brain, heart, liver, pancreas, spleen, stomach, and testes [48]. Its level increases in the blood after physical exercise. Bostrom et al. (2012) observed a 65% increase in irisin concentration in mice running regularly for 3 weeks. An increase in the level of this myokine by two times was also found after 10 weeks of supervised training in the blood of healthy people [7]. Physical exercise is known to increase the level of irisin also in people with metabolic disorders [49]. Nevertheless, irisin level was significantly lower in the serum of urban inhabitants compared to rural subjects, so depending on the activities performed at residential place. In general, irisin concentration is lower in sedentary rather than in active individuals, where the concentration of irisin was found to be about 4.3 ng/mL in the blood of the active subjects and descend to 3.6 ng/mL in sedentary ones [50]. Moreover, some data suggested that the type of physical activity is important in determining irisin level, because irisin induction was noted after resistance training and high-intensity exercise, but not after endurance exercise [51,52].

The circulating level of irisin is related also to the phenotype of different disease. For example, modulation of irisin concentrations were described in obese subjects, in patients with type 2 diabetes (T2DM) [53], chronic renal disease (CKD) [54], and hypothyroidism [55].

The first discovered function of irisin is the ability to induce changes in adipose tissue. Irisin and PGC-1α regulate the expression of UCP-1 and thermogenesis in brown adipose tissue (BAT). As a consequence, the metabolism of glucose and lipids are driven through the increase in energy consumption [56]. Irisin also affects the functioning of white adipose tissue (WAT) by a process named “browning,” where irisin increased the expression of UCP-1 in mature fat cells allowing the conversion of the WAT phenotype in the BAT phenotype. The result is the formation of the beige/brite adipose tissue, the third type of adipose tissue. The treatment of WAT preadipocytes with irisin inhibited adipogenesis and effectively reduced the formation of new adipocytes [57].

Other effects of irisin involved glucose homeostasis and insulin sensitivity improvement in IR, acting on adipose tissue, SkM, liver, and pancreatic β cells [58]. Obesity is usually accompanied by IR, type 2 diabetes, and cardiovascular diseases [59]. Significant changes in the irisin levels have been described in the blood of people affected by obesity. Previous studies suggested that irisin circulating level is reduced in obese subjects [53], on the other hand, more recent studies showed opposite results [60,61]. A state of irisin resistance may be reported during the course of obesity development, which could explain the elevated levels of irisin in these subjects [60].

It was found that the decrease in irisin is associated with an increased risk of presenting metabolic syndrome and hyperglycemia in obese adults, considering it to be protective against IR because it shows negative associations with fasting insulin and glycosylated hemoglobin [62]. To this end, Al-Daghri et al. observed negative association between irisin and fasting glucose and HOMA-IR in school-age Saudi boys and girls [63]. Conversely, there are positive associations between irisin and insulin concentration, fasting glucose, and HOMA-IR [64,65,66,67]. Fukushima et al. analyzed a positive correlation on obese men and adult women [66], while Pardo et al. determined a correlation in women with anorexia nervosa, normal weight, and obesity [65]. Park et al., in a study with people with metabolic syndrome and healthy control, suggested that people with metabolic syndrome had higher concentrations of irisin, associating increased irisin with a greater amount of fat mass in obesity condition, in addition to the possible resistance to irisin or its compensatory role [67]. Nevertheless, it has been suggested that low level of irisin could be due to the reduction in sensitivity to insulin and lipid and glycolytic metabolism [64].

Irisin also exerts anti-inflammatory and antiapoptotic effects on cardiomyocytes and hepatocytes, protecting these cells from ischemia-reperfusion injury [68,69]. These effects are mediated by the induction of autophagy and have been described in more details in Section 7. Irisin has also favorable effect on brain, increasing the proliferation of mouse hippocampal neuronal cells [70], and reducing oxidative stress-induced neuronal damage by inhibiting the secretion of proinflammatory cytokines [71]. In mouse model, the FNDC5/irisin system is fundamental for long-term potentiation and memory in the hippocampal region, the expression of FNDC5/irisin is indeed necessary to establish synaptic plasticity and memory. In human, a study on adult volunteers at risk of dementia submitted to 10 weeks of physical training, showed a positive correlation between cognition/episodic memory and serum irisin [72]. In addition to serotonin, irisin may contribute to the antidepressant effect of exercise, most likely associated with the activation of the PGC-1α/BDNF pathway [73]. At bone level, irisin acts as a link between SkM and this tissue, increasing the mass and strength of the cortical bones and positively modifying their geometry by reducing the secretion of osteoblast inhibitors and causing the expression of bone-specific genes [74]. Nevertheless, irisin mediates the positive anti-inflammatory effect induced by regular and moderate physical activity on the functioning of the immune system reducing systemic inflammation [75] and protecting from the development of diseases associated with chronic inflammation.

## 5. Exercise and Autophagy

Physical exercise exerts beneficial effect on adults’ health and is actually considered as a nondrug therapy against multiple diseases such as metabolic diseases, cardiovascular diseases, and neurological diseases [11]. Regular physical activity may be an effective autophagy inducer [76]. Accordingly, exercise has been shown to augment acute autophagic activity in SkM and in several other tissues. These include adipose tissue, heart, liver, pancreatic β cells, peripheral blood mononuclear cells (PBMCs), and brain [77,78,79,80].

In 1984, Salminen and Vihko, first discovered the induction of autophagy caused by strenuous exercise in SkM of mice [81]. To follow, autophagic degradation of proteins in liver induced by endurance exercise is reported by Dohm in 1987 [82]. In 2011, Grumati and coworkers, suggested that ultraendurance exercise can activate autophagy in SkM, which is confirmed by the increased conversion of LC3I to LC3II [83].

In SkM, the delivery of a substitute energy source is the main function of autophagy during exercise, this tissue accounts for the most substantial proportion of the body in relation to the increasing demand for oxygen and glucose. In this environment characterized by low sugar and low oxygen, autophagy plays a vital role in preventing excessive fatigue and damage of SkM and permits to this tissue to achieve the adaptation toward exercise. Together, autophagy contributes to improve and stabilize metabolism, by strengthening the removal and recycling of damaged or denatured proteins and/or organelles in SkM [84,85]. Other cellular tasks induced by exercise may promote autophagy in muscle during exercise, including elevated mitochondrial damage and respiration, high concentrations of reactive oxygen species (ROS), the immune response, counting the presence of diverse cytokines, and alterations in calcium and nicotinamide adenine dinucleotide (NAD^+^) [86].

The exercise acts to initiate autophagy in SkM through AMPK and SIRT1 that are, respectively, sensitive to alterations in AMP and NAD^+^. AMPK and SIRT1 both activate the transcription factors FOXO1 and FOXO3 to upregulate expression of Atgs genes by the inhibition of mTORC1 and increasing PGC-1α activity [86], while AMPK also initiates autophagosome formation via ULK1 [63,72]. mTORC1 has also been implicated in regulating autophagy activity through transcription factor EB (TFEB) localization, whereas physical exercise is one of influencing factor. At rest condition, mTORC1 confines TFEB in the cytosol after its phosphorylation on the lysosomal surface. In the upcoming exercise, mTORC1 disassociates from lysosome determining the dephosphorylation of TFEB that translocates to the nucleus where it then activates the transcription of Atg genes and proteins [87].

In an elegant work, using a mutant rodent model that inhibits exercise-induced autophagy, He et al. (2012) reported that autophagy-deficient mice demonstrated impaired glucose uptake, glucose transporter 4 translocation, and AMPK activation during acute exercise. Other data suggest that autophagy confers additional benefits of exercise as well, mitochondrial biogenesis, improved endurance, and angiogenesis [12,77,83,88].

The exercise-induced autophagy in SkM is widely studied, but several data showed enhanced autophagic activity in other tissues, thus demonstrating that physical exercise is capable of inducing a systemic autophagic reaction. Endurance exercise improves autophagy activity in liver, heart, adipose tissue, and pancreatic β cells of wild type mice, but not in autophagy-deficient mutant mice [77]. Nevertheless, acute exercise increased autophagic flux in the brain, at anterior cerebral cortex [12]. In myocardium, Ogura et al. (2011) showed, for the first time, that exercise causes a biphasic change in autophagy with an initial reduction in LC3II level following exercise, and a subsequent upregulation for 1 h thereafter, which might be associated with regulation of mTOR [89]. More recently, Li and coworkers suggested that during exercise and up to 24 h after, some mitochondrial autophagy-related proteins and autophagy flux were upregulated in myocardium of a mouse model [90]. In mouse hepatocytes following a high-fat diet, the expression of many Atg proteins were recovered in response to endurance exercise (10 weeks) and were related to lipogenic gene expression and reduced lipid content [80]. Physical exercise also modulates autophagy in PBMCs. For example, 1 h of exercise in a 30 °C warm environment increased autophagy in these cells [78]. Additionally, Mejias-Pena et al., in two recent works, showed that 8 weeks of aerobic training and resistance training upregulated the expression of several Atg proteins and autophagy activity in PBMCs in older adults [91,92].

These evidences suggest that exercise-mediated autophagy could interfere with several pathology in which autophagy dysregulation is implicated, such as T2DM [93], neurodegeneration [94], cardiomyopathy [95], and chronic inflammation [96] while sustaining muscle quality and function [97].

Although the exercise-induced autophagy is regulated by duration and intensity [98], an amount of exercise dose has yet to be established. Actually, evolving data from rodent and human models support the view that chronic exercise improves autophagy activity [17,80,91,92,99,100,101]. In mice, it has been reported that markers of autophagy activity (including LC3, Atg7, Beclin1, and FOXO3) were upregulated in SkM after 4 and 8 weeks of endurance training [17,99]. Lifelong exercise produced higher SkM expression of mRNA of Atg7 and Atg9 in mice. Considering humans, one study is available suggesting that older, overweight women showed significant increase in LC3, Atg7, and FOXO3 mRNA expression after 6 months of walking at moderate intensity and resistance training. These women were also accompanied by improvements in body composition [100,101]. The effects of exercise on autophagy in tissues are summarized in Figure 3.

## 6. Myokines and Autophagy

The exercise induces autophagy in SkM and in other tissues, and several beneficial effects of physical activity on adults’ health can be attributed to its property as autophagy inducer [76]. A great amount of data suggested that SkM can regulate both systemic physiology and aging via the release of myokines. Exercise has been shown to promote the muscle secretion of the myokines such as IL-4, IL-6, IL-7, IL-15, decorin, irisin, LIF, metrnl, myonectin, and myostatin, all of which could induce or inhibit autophagy throughout muscle and possibly nonmuscle tissue [102]. In general, the role of myokines in regulating autophagy has not been extensively studied. In this section, we consider the most relevant evidences describing the effect of myokines on autophagy in several tissues. The role of irisin in modulating autophagy is treated in detail in the following section.

### 6.1. Myostatin

Myostatin is an important regulator of SkM mass that drives the reduction in muscular mass by inducing the loss of proteins. To this end, myostatin acts promoting the activity of the ubiquitin–proteasome and the autophagy-lysosome systems. The relationship between myostatin and autophagy has been considered in numerous previous studies. First of all, Lee et al. (2011) demonstrated that myostatin promotes autophagy in myoblast C2C12 cells, in the absence of horse serum in the medium, showing an increase in LC3II and autophagosome number in these cells incubated with myostatin [103]. The in vitro myostatin-mediated trout myotube atrophy, is associated with both the ubiquitin–proteasome and the autophagy–lysosome systems [104]. In C2C12 cells differentiated in myotubes, the TNF-α mediated activation of myostatin, upregulated the expression of gene encoding for proteasome subunits and genes related to autophagy (e.g., MAFbx, MuRF1, LC3II), inducing atrophy. The authors summarized that myostatin mediated muscle catabolism via coordinate activation of the autophagy and the ubiquitin-proteasome systems, in particular, treating cells with the PI3K inhibitor LY294002, and FOXO3a was inhibited upstream for the prevention of the induction of the abovementioned autophagy-related genes [105].

More recently, Manfredi and coworkers, studied the relationship between myostatin in primary cultures of neonatal cardiac myocytes from rats. They showed that cardiomyocytes incubated with myostatin have an increase in proteolysis by activating autophagy and the ubiquitin proteasome system, and a decrease in protein synthesis by decreasing P70S6K. In parallel, they considered the Extensor digitorum longus muscles from mice and C2C12 cells, in which myostatin induced an increase in proteolysis with an increase in Atrogin-1, MuRF1, and LC3 genes [106]. Furthermore, myostatin, acting through the GRK2 kinase, leads to impaired mitochondrial respiration, which was associated with mitochondrial fragmentation, enhanced autophagic flux, upregulation of LC3II and p62, and reduced mitochondrial content in C2C12 cells [107].

Another study used Apc^Min/+^ mice, an established model of colorectal cancer and cachexia, and mice engrafted with Lewis lung carcinoma cells, evidencing that inactivation of the myostatin gene prevented the loss of SkM mass that characterize these models. At molecular level, the inhibition of myostatin loss inhibited the expression of atrophy-related genes, MuRF1 and MAFbx/Atrogin-1, reducing the activation of muscle fiber proteolytic pathways. These results reveal an essential role for myostatin in the pathogenesis of cancer cachexia [108].

### 6.2. Decorin

Decorin is the first soluble proteoglycan known to influence autophagy in normal endothelium [109]. It has been shown to inhibit migration of glioma cells by activating autophagy and inhibiting TGFβ signaling [110]. This myokine induces autophagy in the intestinal cells and reduces the symptoms of inflammatory bowel disease in mice [111]. In endothelium, decorin interacts with the VEGFR2 ectodomain and triggers the proautophagic AMPKα/Vps34 signaling pathway activating the AMPKα at Thr172. This event stimulates the synthesis of the transcription factor Peg3 (paternally expressed gene 3) and leads to the autophagic process. In turn, this stimulates the synthesis of Peg3 (Paternally expressed gene 3), which is directly bound to Beclin1 and localized to autophagosomes. By binding to VEGFR2, decorin suppresses VEGFA levels and blocked the mTOR pathway similarly to rapamycin. At the end, the decorin/VEGFR2/Peg3 signaling cascade leads to induction of the proautophagic genes BECN1 and MAP1LC3A (gene encoding LC3) [109,112,113,114].

More recent findings suggested that in the presence of decorin, Peg3 induced the TFEB expression and nuclear translocation. When VEGFR2/AMPKα signaling is blocked by Dorsomorphin, the decorin-mediated TFEB induction is abolished. As confirmed, the decorin-mediated induction of autophagosome positive for Beclin1 and LC3 is stopped when TFEB is silenced. Of note, decorin is actually the only proteoglycan that is induced by an autophagic stimulus. In detail, in mouse cardiac tissue, fast diet inhibited the canonical pathway of mTOR and induced expression of decorin at mRNA and protein level. Nevertheless, knockout mice for decorin are characterized by impaired autophagy level at cardiac level [115].

### 6.3. IL-4

The majority of evidences on IL-4 and autophagy derived from studies on immune cells, in relation to the study of microglial cells activation and its response during the pathological condition of asthma, or in differentiation and antigen processing. In in vitro model of microglial cell line (N9), IL-4 downregulated the constitutive autophagy and downregulated the protein levels of Beclin1, involved in the induction of autophagy [116]. More recently, Tang et al. (2019) suggested opposite results and suggested that dose and time of IL-4 stimulation, are important in regulating autophagy process in another microglial cell line (BV2). IL-4 induced microglial autophagic flux, most significantly at 20 ng/mL for 48 h. The authors suggested that IL-4 may play a beneficial role against Alzheimer’s disease because pretreating microglia with IL-4 induced autophagy that partly contribute to increase in amyloid-β uptake and degradation [117].

Other evidences on IL-4 and autophagy came from the study of allergic asthma. Asthma-prone mice were generally characterized by higher level of autophagy in pulmonary B cells. In accordance, when autophagy is blocked in these cells, the pathological symptoms were reduced in asthma-prone mice. Contextually, IL-4 play a key role for autophagy induction in vivo and in vitro, sustaining B cell survival and enhancing antigen presentation. At molecular level, IL-4-induced autophagy was not dependent by mTOR, but involved the JAK signaling, and was dependent by PI3K pathway [118]. Dickinson et al. (2018) studied the effect of IL-4 stimulation on autophagy in primary-culture human tracheobronchial epithelial cells, showing that prolonged 7-day treatment with IL-4 increased autophagosome formation and degradation, determined by LC3BII protein levels, immunostaining, and intracellular superoxide levels, while brief activation had no effect. The authors concluded that IL-4-mediated increase in autophagy was dependent on changes in epithelial cell function that occur after chronic cytokine stimulation, but was not dependent on immediate STAT6 signal transduction pathways that occurs after brief stimulation [119].

The increase in autophagy flux is also required in dendritic cells (DCs) during differentiation and antigen presentation. To this end, the stimulation with IL-4 regulated mTORC1 and induced RUFY4 induction, which ultimately induce the degradation of LC3, the formation of autophagosome positive for Syntaxin 17, and lysosome binding. In IL-4–treated DCs and in RUFY4-expressing cells, the enhancement of autophagy supports endogenous antigen presentation by MHC II and allows host control of Brucella abortus replication. RUFY4 is a novel regulator that augments autophagy flux and, when overexpressed, induces drastic membrane redistribution and strongly tethers lysosomes [120,121].

### 6.4. IL-6

During the last decade, it has been reported that IL-6 regulates the autophagic process through both inhibitory and stimulating effect. Some studies showed that IL-6 acts as a contributing factor of autophagy in cancers, suggesting that the IL-6/p-STAT3 pathway is a positive regulator of autophagy in pancreatic cancer [122] and in prostate cancer [123]. More recently, Xue et al. (2016) described a role for IL-6 in initiating autophagy also in glioblastoma [124].

The first evidence in nontumor tissue for IL-6 in modulating autophagy, arrived from Kimura et al. (2010), reported an involvement of IL-6/STAT3 signaling pathway in inhibiting autophagy in renal tubular epithelial cells, but without any further research on downstream key signal molecules about the autophagic process [125]. Following study on neuronal cell line (SHSY5Y) showed that IL-6 promotes the formation of autophagosomes containing cellular debris, suggesting that this effect is dependent on the activation of ERK, but not on p38 pathway [126]. Conversely, Quin et al. (2015) showed that IL-6 exerted antiautophagic effects in vitro on monocytic cell line. In starved U937 cells, the addition of IL-6 significantly activated the phosphorylation level of STAT3 at Tyr705 and reduced the protein levels of LC3II and Beclin1. Knockdown or overexpression of Beclin1 indicated the involvement of PI3K in IL-6-mediated inhibition of autophagy in these cells [127]. Considering the vascular smooth muscle cells (VSMCs), An et al. (2017) demonstrated that stimulation with IL-6 enhanced autophagy and downregulated the expression of VSMCs contractile proteins α-SMA and SM22α. They suggested that the autophagy-related 4B cysteine peptidase (Atg4B) plays an important role in IL-6-induced autophagy upregulation. Atg4B knockdown, indeed, blocked IL-6-induced autophagy and α-SMA and SM22α degradation [128]. The role of IL-6 on autophagy has been well investigated in the β islet cells of pancreas. In this tissue, IL-6 stimulates autophagy inhibiting mTORC1 and activating STAT3, Akt, and the autophagy enzyme GABARAPL1 and protecting cells from apoptosis under pro-inflammatory stimuli [129]. Nevertheless, IL-6 couples autophagy to antioxidant response and reduces ROS in mice β-cells and human islets. IL-6-driven ROS reduction is associated with an increase in the master antioxidant factor NRF2, which stimulates autophagy, and promotes survival under diabetogenic conditions [130]. Recently, the role of IL-6 on liver autophagy has been investigated under physical exercise condition, showing that IL-6 KO mice have inhibited some hepatic autophagy markers at baseline (i.e., ULK1 and Atg5). The acute exhaustive exercise reduced mRNA levels of genes related to the autophagy pathway (e.g., FOXO1a) in the livers of WT mice; however, these effects were attenuated for the IL-6 KO group, highlighting a new function of this cytokine [131].

### 6.5. IL-7

Some studies showed the direct relationship of IL-7 in decreasing autophagy in different cancer and normal cell types. IL-7 is an inhibitor of autophagy in T cell. For instance, when the IL-7-dependent T-cell line D1 were deprived of IL-7, they had an enhanced ratio of LC3II to LC3I, suggesting an increased number of autophagosomes. In the same cells, the levels of p62 decline when IL-7 was withdrawn, indicative of increased autophagic degradation [132]. Zhu et al. (2018), considering macrophages, suggested that *Schistosoma japonicum* (*S. japonicum*) egg antigen (SEA) triggered autophagy limiting the development of pathology in host liver. In this context, the stimulation by *S. japonicum* of the receptor for IL-7 (IL-7R/CD127) induced IL-7, strongly reduced SEA-triggered macrophage autophagy, leading to an enhanced liver pathology. In addition, the liver pathology was significantly repressed when macrophage autophagy was increased by anti-CD127 blocking antibody treatment or anti-IL-7 neutralizing antibody. This effect of IL-7 was allowed through activation of AMPK [133]. More recently, Jian et al. (2019), purposed the involvement of IL-7 in blocking autophagy in human lung cancer cells (A549 and H460 cell lines). IL-7 and its receptor IL-7R, are able to induce PI3K/Akt/mTOR signaling pathway via downregulating Beclin1. In parallel, they showed, by in situ analysis of human NSCLC, that overexpression of IL-7/IL-7R and mTOR plays an important role in decreasing lung cancer cell autophagy and increases tumor development [134].

### 6.6. IL-15

The cytokine/myokine IL-15 mediates development and survival of immune cells, including T cell and natural killer cell (NK). The main findings describing its role on the autophagic system have been obtained from recent studies on these cell types. The first study addressing on the IL-15 autophagy modulation has been performed by Wang and coworkers (2016). They showed that elevated level of autophagy appears in the stage of immature NK (iNK) and is required for cell development, whereas defective autophagy results in accumulation of ROS from damaged mitochondria and cellular apoptosis. The inhibition of IL-15 signaling abrogated the phosphorylation of FOXO1 and its consequent translocation to the cytoplasm of iNKs, leading to autophagy inhibition. At rest condition, the phosphorylated form of FOXO1 is situated at the autophagosome level in the cytoplasm of these cells, where it interacts with Atg7, and leads to autophagy induction. When FOXO1 is deficient or is an inactive FOXO1^AAA^ mutant, autophagy initiation was abrogated in iNKs and NK cell development and viral clearance is impaired [135].

In addition, IL-15 induces autophagy in natural killer T (NKT) cells with a mechanism that involves the TBK-binding protein 1 (Tbkbp1). Tbkbp1 facilitates activation of ULK1 antagonizing the inhibitory action of mTORC1. The suppression of Tbkbp1 expression reduces the level of autophagy in NKT cell, stimulated with IL-15, and is associated to reduced NKT cell survival after mitochondrial dysfunction, ROS production, and decreased Bcl2 expression [136]. Other evidences on T cells suggested that IL-15 prolongs T-cells survival and improves their memory when transferred in irradiated lymphopenic mice. In details, T-cell survival mediated by IL-15 signaling involves the activation of STAT5/Bcl2 pathway and enhances T-cell memory formation via IL-15-mediated activation of the FOXO/eomesodermin memory, ULK1/Atg7 autophagy pathways, and mitochondrial remodeling [137].

The intrahepatic lymphocytes were characterized by increased rate of basal autophagy, mainly the CD8^+^ T cells resident in liver. In these cells, the higher level of autophagy is positively correlated to proliferation, cytotoxicity, and cytokines production. On the other hand, reduction or blockage of the induction of autophagy leads to higher level of T cells characterized by exhausted phenotype, and inhibits the tissue-residence programming. In CD8^+^ T cells, the induction of autophagy by IL-15 was paralleled by the expression of markers (CD69^+^ and CD103^+^) for tissue retention, was IL-15 dose-dependent, and was blocked when T cells were treated with anti-IL-15 monoclonal antibody or by the addition of recombinant human IL-15Ra [138].

### 6.7. Other Myokines

LIF is a 20-kDa myosin-related pleiotropic cytokine secreted as well as by SkM, heart, and nervous system. This myokine can pass through the blood–brain barrier into the nervous system, where it is mainly involved in the development of astrocytes and survival of oligodendrocytes. In HT-22 mouse hippocampal cells and primary hippocampal cells, LIF activated Akt/mTOR/S6K, ERK/signal transducer, and STAT pathways and reduced autophagy, as shown by LC3II diminished expression. The result was the final increase in the neuronal activation marker c-fos and increase in cell survival. Furthermore, suppression of the autophagy marker, LC3II, by LIF was observed in a Drosophila model of Alzheimer’s disease [139].

Other evidence is available for myonectin (CTRP15), as inhibitor of the autophagy process under starvation. In mouse liver and cultured rat H4IIE hepatocytes, treatment with recombinant myonectin inhibited the induction of autophagy mediated by starvation. The process was supported by the inhibition autophagosome formation, p62 degradation, and expression of critical autophagy-related genes (i.e., Atg7 and Atg12). The inhibition of PI3K/Akt/mTOR pathway annulled the suppression of autophagy exerted by myonectin in cultured hepatocytes, suggesting that this pathway has a key role in the reduction in proteins degradation [140].

Metrnl is a recently discovered protein that is secreted by SkM, but also highly expressed in the intestinal epithelium. The role of Metrnl has been studied in intestinal epithelial cell-specific Metrnl knockout mice (Metrnl^−/−^). In this model, the administration of a pro-inflammatory stimulus such as 3% dextran sodium sulfate (DSS) drinking water, induced colitis’ symptoms that were more severe in Metrnl^−/−^ mice than in WT mice. Of note, Metrnl^−/−^ mice were defective in DSS induction of autophagy in the colon, as suggested by higher level of p62 and lower levels of Beclin1 and LC3II/I. In parallel, the intestinal epithelial cells deficient for Metrnl, were characterized by an increase in phosphorylated form of mTOR and p70S6K and reduction in AMPK, compared with their WT comrades. This suggested that treatment with DSS activated the autophagy-related pathway AMPK/mTOR/p70S6K during colitis induction [141].

## 7. FNDC5/Irisin and Autophagy

Several studies have successfully showed that autophagy exerts a fundamental role in health and disease. Autophagy acts in degrading several cytoplasmic aggregates of proteins that cause some neurodegenerative diseases, acting on tau protein in various forms of dementia, on mutant huntingtin in Huntington’s disease, and on α-synuclein in Parkinson’s disease. Similarly, autophagy protects against infectious diseases (e.g., *Mycobacterium tuberculosis*). Under these conditions, the autophagy induction showed beneficial effects in animal models by reducing cell death and alleviating the inflammatory state. In addition, the autophagic process has an important role in cancer, whereas it protects against the initiation of carcinogenesis and has a role in solid tumors characterized by low nutrients, allowing the survival of cells. Nonetheless, these processes should be better investigated because they strictly depend on the genetic constitution of the relevant tumor. In addition, autophagy also appears to have roles in metabolic diseases and heart, liver, and kidney diseases among many other pathological and physiological situations [142]. Recent findings on several tissues posed the autophagy as an essential part of the molecular machinery induced by irisin exerting the favorable effects of physical exercise. To follow, we focused on the pathological conditions in which irisin has been candidate as potential beneficial factor by autophagy induction.

### 7.1. Metabolic Disorders

Irisin may be generally considered an option to prevent and treat metabolic disorders. It has effects on glucose homeostasis and insulin sensitivity by promoting energy expenditure, glucose uptake, glycogenolysis, gluconeogenesis reduction, adipogenesis, and lipid accumulation [56]. Irisin has the capacity to activate beneficial changes in adipose tissue that improve muscle activity. Therefore, moderate increases in irisin produces an improvement in IR within the diet induced changes. This property candidates irisin as a potential therapeutic target in metabolic diseases, including metabolic syndrome, T2DM, and nonalcoholic fatty liver disease (NAFLD), in which IR plays a major pathogenic role [143]. Additionally, irisin may promote pancreatic β-cells survival and improve glucose-induced insulin secretion under lipotoxic conditions [58] or high glucose state [144]. Several studies have evaluated the correlation between blood glucose levels and irisin, β-cell function, and IR in nondiabetic subjects. In general, irisin levels positively correlated with blood glucose under fast condition [64,145,146]. Lower irisin levels have been described in drug naïve T2DM patients and in populations with prediabetes [146,147]. Due to these evidences, irisin might be considered for the management of T2DM in the future. Nowadays, some studies have investigated this possibility. First, a study on diabetic mice suggested that metformin administration increases FDNC5 mRNA/protein expression and irisin levels in a way that is independent from AMPK activation [148]. Furthermore, other studies described that administration of irisin increased glycogenesis, reduced gluconeogenesis, and, in summary, improved muscle IR by the AMPK pathway in diabetic and insulin-resistant mice models [149,150].

Recent evidences support the idea that the abovementioned positive effects of irisin on IR might be mediated by autophagy regulation. To this end, Ye et al. (2019) in an in vitro model shows that irisin ameliorates mitochondrial respiration and function, stimulating autophagy by induction of PGC-1α. As a result, C2C12 myoblast cells increase glucose uptake via the p38-mitogen-activated protein kinase (MAPK)-PGC-1α pathway. These data support the evidences that irisin can enhance mitochondrial function and may act on p38-MAPK-PGC-1α pathway activation, posing the bases for therapeutic potential of irisin in treating IR. In this study, treatment with irisin at 5 nM significantly augmented the autophagic rate, as ascertained by increased level of the autophagosome marker LC3, and increased degradation of p62. The effect was enhanced when the irisin concentration was raised to 15 nM, suggesting that in this context, irisin could act in a dose-dependent manner. In detail, the transport of LC3 to the cytoplasm from the nucleus to form autophagosome around the nucleus was increased by irisin, accompanied by significant increase in the conversion of LC3I into LC3II compared with the positive control (cell treated with 3-MA alone). Nevertheless, the transport of the marker of autophagy induction TFEB was increased by irisin in a dose-dependent manner. This process was dependent by PGC-1α, whereas the nuclear translocation of TFEB was annulled after the knockdown of PGC-1α. Irisin-induced reversion of IR in C2C12 cells is dependent on induction of autophagy via PGC-1α [151].

The irisin-mediated induction of autophagy is responsible also of part of the protective function of metformin on pancreatic β cells in alleviating IR. In an in vitro model of INS-1 cells, high glucose environment induced lower expression of irisin and proteins associated with AMPK/SIRT1/PGC-1α pathway and autophagy, lower proliferation rate, lower proliferation, and higher expression of proteins associated with apoptosis. Treatment with metformin significantly rescued lower cells proliferation, glucose-stimulated insulin secretion, and expression of proteins associated with AMPK/SIRT1/PGC-1α signal pathway and autophagy, as well as irisin, and suppressed apoptosis in high-glucose environment. Under these conditions, autophagy inhibitor chloroquine and SIRT1 inhibitor Ex527 can block the abovementioned functions of metformin. Therefore, irisin, through induction of autophagy, determines the metformin-mediated INS-1 cells proliferation, enhances glucose-stimulated insulin secretion, and suppresses apoptosis by activating AMPK/SIRT1/PGC-1α signal pathway in high-glucose environment [152].

Irisin expression has been detected in hepatocytes immunohistochemically [153], and liver expresses FNDC5 mRNA and protein, albeit in lower amounts than the adipose tissue and SkM [64]. Liver and hepatocyte are targets of irisin. A study based on single-photon emission computed tomography detected high radioactivity of 125I-labeled irisin in the mouse liver. Furthermore, after treatment with irisin, it was successfully detected in mouse primary hepatocytes, and in AML12 cells (mouse hepatic cells), suggesting the presence of an unknown irisin receptor in hepatocytes that has not yet been studied. Studies on animal models suggested that irisin is involved in improvement of glucose homeostasis acting on lipid and glucose metabolism in liver. In general, irisin increases hepatic glycogenesis and reduces gluconeogenesis in mouse and human primary hepatocytes, in HepG2, a human hepatocellular carcinoma cells, and in diabetic mice. At molecular level, these properties were driven by the activation of PI3K/Akt and AMPK [150,154,155]. In ob/ob mice, a leptin-deficient model exerting severe steatosis, the increase in serum irisin was elicited by the administration of adenovirus carrying human FNDC5 cDNA, in turn determining the suppression of hepatic gluconeogenic and lipogenic enzymes, improving IR, and decreasing hepatic triglyceride in this mouse model [155]. As confirmed by studies on irisin transgenic mice, this myokine reduced the oxidative damage induced by higher level of H_2_O_2_-mediated reactive oxygen species and may also improve the survival of the hepatocytes under oxidative stress condition, which are strictly related to the pathogenesis of NAFLD [156,157].

In this context, autophagy induction mediates some of the beneficial effects of irisin on liver. Under fast condition, FNDC5^−/−^ mice were characterized by liver reduction in autophagy and fatty acid oxidation (FAO). The knockout mice showed severe steatosis and enhanced lipogenesis when compared to WT mice. Accordingly, primary culture of hepatocytes from FNDC5^−/−^ mice had lower response to energy deprivation in terms of autophagy induction, AMPK activity, and FAO. Treating cells with the AMPK activator 5-aminoimidazole-4-carboxamide ribonucleotide (AICAR), significantly recovered autophagy rate and FAO. Nevertheless, the treatment with the mTORC1 inhibitor, rapamycin, reduced lipogenesis and steatosis, enhancing autophagy and FAO in the same model.

In obese mice, the deficiency of FNDC5 aggravated autophagy impairment, hyperlipemia, hepatic FAO and lipogenesis, and lipid accumulation. The administration of recombinant FNDC5 repaired the attenuated autophagy in FNDC5^−/−^ hepatocytes and in palmitate-induced steatosis, stimulating gene expression of targets related to autophagy and FAO. In addition, the overexpression of FNDC5 prevented autophagy impairment, leading to better response to hyperlipemia, hepatic FAO, hepatic lipogenesis, and lipid accumulation. The authors concluded that higher level of FNDC5 counteracting impaired autophagy and FAO in the liver could prevent the high-fat diet-induced hyperlipemia and hepatic lipid accumulation, whereas the deficiency of FNDC5, via the AMPK/mTOR pathway, could impair autophagy and FAO and enhance lipogenesis [158].

### 7.2. Myocardial Ischemia/Reperfusion Injury

A considerable amount of the circulating level of irisin in animals has origin from the myocardium secretion, being one of the largest muscles in the organism [159]. The secretion of irisin depends from the heart condition. In particular, ischemic conditions, as coronary artery disease (CAD), and myocardial infarction (MI) significantly modulate the secretion of irisin [160]. Specific stress or damage in cardiomyocytes could directly induce irisin secretion, nevertheless adverse lipid profile characterizing people that suffer from these conditions could modulate irisin expression and release. Initially, it was sustained that CAD/MI could induce higher irisin level directly from damaged cardiomyocytes. More recently, a study on rat model suggested that in isoproterenol-induced MI, the irisin expression and release were reduced and the circulating levels were negatively associated with troponin and creatine phosphokinase-myocardial band isoenzyme (CK-MB), hallmark markers of cardiac damage [161]. In human, such evidences have been observed, whereas patients with CAD and MI showed lower level of irisin than relative controls [158,159,160,162]. In animal model, irisin administration revealed positive cardiovascular effects. Irisin preserved the function of mitochondria, determining beneficial effect by inhibiting apoptosis in cardiomyocytes under ischemic condition [163]. Aronis et al. (2015) were the first to investigate the relation between cardiovascular diseases and irisin. They investigated the possible role of irisin as a predictor of acute coronary syndrome in healthy people, suggesting that irisin could predict adverse coronary events in patients with coronary artery diseases under treatment with percutaneous interventions. Thus, decreased level of circulating irisin in this population have a 12-months free survival rate following percutaneous coronary intervention [164].

Ischemia and cardiomyocyte death are the main characteristics of the MI. Under ischemic conditions, at cellular level, the mitochondria have a fundamental role in counteracting cellular damage and sustaining cardiomyocyte viability. Under these conditions, autophagy is a key mechanism to maintain mitochondrial structure and function through the elimination of the damaged mitochondria. Xin et al. (2020) studied the dynamin-like GTPase optic atrophy 1 (Opa1) in infarcted hearts. They suggested that the expression of Opa1 was downregulated in vitro in hypoxia-treated cardiomyocytes and in vivo after MI. Interestingly, they found that autophagy was enhanced by Opa1 overexpression, improving cell viability and protecting cardiomyocytes against hypoxia-induced damage. Treatment with irisin was capable to mediate the Opa1-induced autophagy and protect cardiomyocytes from further damage following MI. In addition, treatment with irisin restored the mitochondrial membrane potential, a marker of mitochondrial function, and abrogated the downregulation of the levels of the mitochondrial respiratory complex, resulting in an increase in ATP levels in response to hypoxia. In addition, irisin improved the level of components of the antioxidative system, comprising glutathione (GSH), superoxide dismutase (SOD), and glutathione peroxidase (GPX), that were generally reduced during ischemia. The protective effect of irisin was annulled when Opa1 was silenced, resulting in cardiomyocytes’ higher level of mitochondrial dysfunction, oxidative stress, and inflammation. Opa1 knockdown also inhibited the restoration of the mitochondrial membrane potential and ATP level mediated by irisin in cardiomyocytes [69].

### 7.3. Myocardial Hypertrophy

Deficiency in autophagic flux is considered a pathology feature of hypertrophic hearts, leading to heart failure after impaired cardiac remodeling. In two recent studies, Li and coworkers aimed to investigate the cardioprotective role and mechanisms of irisin in cardiac hypertrophy and remodeling under different stimuli inducing pressure overload. In a first study, 4 weeks of transverse aortic constriction (TAC) alone or combined with intraperitoneal injection of chloroquine diphosphate (CQ) were administered to adult male wild-type, mouse-FNDC5 knockout and FNDC5 transgenic mice. They described that irisin exerted protection against cardiomyocytes hypertrophy induced by angiotensin II (Ang II) or phenylephrine (PE), which was comparable to exogenous FNDC5 overexpression that attenuated the TAC-induced hypertrophic damage in the heart. Conversely, the ablation of endogenous FNDC5 aggravated the hypertrophic damage. The cardiomyocytes treated with Ang II- or PE and the TAC-treated myocardium were characterized by impaired autophagy flux and accumulation of autophagosomes. This condition was worsened under irisin deficiency condition, showing reduced autophagy and higher autophagy flux failure. In accordance, supplementation or overexpression of irisin improved autophagy flux. The process was abrogated by the inhibitors of autophagy CQ, 3-MA, and siRNA for Atg5. The inhibition of AMPK and ULK1, respectively, with compound C and SBI-0206965, reversed the induction of both autophagy and autophagy flux and annulled the protective role of irisin against cardiomyocytes hypertrophic injury. In detail, irisin activated AMPK and ULK1 at Ser555 (no Ser757) in cultured cardiomyocytes and hypertrophic hearts. However, its effect did not include Akt and MAPK family members and did not affect the mTOR-S6K axis [165]. In a more recent work, they investigated the effect of angiotensin II treatment in response to pressure overload, whereas it has been well established that angiotensin II significantly contributes to cardiac hypertrophy. The expression of irisin was reduced by angiotensin II in cardiomyocytes, in parallel with induction of a larger cross-sectional area and higher number of apoptotic cells. The supplementation with irisin reduced cardiomyocytes apoptosis and increased autophagy, as ascertained by analysis of higher level of LC3II expression and decreased p62 expression, meanwhile, the autophagy inhibitor 3-MA reversed the protection exerted by irisin. In summary, the authors concluded that in animal model, overexpression of irisin alleviated myocardial hypertrophy induced by pressure overload reducing cardiomyocytes apoptosis and that irisin protection against pressure overload-induced cardiac hypertrophy is due to induction of protective autophagy and autophagy flux via activating AMPK-ULK1 signaling [166].

### 7.4. Hepatic Ischemia/Reperfusion Injury

The reparative capacity during ischemia-reperfusion injury is decreased in aged liver. A recent study clarified the role of irisin and autophagy during hepatic ischemia-reperfusion in the elderly. During hepatic ischemia-reperfusion, old rats (22 months) exhibited lower level of irisin expression and autophagy rate than young rats (3 months). These differences were paralleled by more severe liver injury, lower telomerase activity, and mitochondrial function. Old rats treated with exogenous recombinant irisin showed higher level of LC3BII and p62 proteins, accompanied by higher number of autophagosomes when compared to control group treated with vehicle alone. In accordance, young rats treated with irisin-neutralizing antibody exhibited significant reduction in performing autophagy after ischemia-reperfusion. In aged hepatocytes, irisin increased the telomerase activity improving mitochondrial function. This finding was confirmed by treating aged hepatocytes with the selective inhibitor of telomerase activity, BIBP1532, that abrogated the beneficial effect of irisin during hypoxia and reoxygenation. The irisin-mediated induction of telomerase activity and autophagy rate were inhibited by anisomycin, a selective inhibitor of JNK, suggesting the involvement of this kinase in the process in aged primary hepatocyte. In this model, administration of recombinant irisin also reduced the level of oxidative stress, inflammation, apoptosis, and as a consequence, the liver injury. In summary, irisin could act as a beneficial factor during hepatic ischemia-reperfusion through autophagy in aged liver [68].

### 7.5. Chronic Kidney Disease

The reduction in irisin level has been detected in stage 5 of CKD, although the reduction in muscle mass and/or the significant reduction in renal function may affect irisin levels. Nevertheless, substantial reduction was observed in diabetic nephropathy patients [167]. Recently, reduction in FNDC5 expression was significantly associated to enhanced SkM autophagy and atrophy under urotensin II (UII) induction in vivo in a mouse model of CKD and in vitro in C2C12 cells. In 5/6 nephrectomy mice, the downregulated FNDC5 expression, and upregulated expression of the autophagy markers LC3II and p62 in SkM tissue, was paralleled by the upregulation of UII expression, supporting the reduction in muscle weight and SkM cross-sectional area in the two posterior limbs. In the same model, the downregulation of FNDC5 expression and the upregulation of autophagy markers, as well as the decrease of SkM cross-sectional area and SkM weight, were inhibited in UT knockdown mice. In accordance, UII treatment inhibited in vitro the expression of FNDC5, induced autophagy markers upregulation, and decreased myotubes diameter after 48–72 h of exposure. When cells were treated with UT-specific siRNA for UII receptor gene, the downregulation of FNDC5 and the induction of the autophagy markers upregulation, UTII mediated, were abrogated [168].

## 8. Conclusions

The health-promoting effects of physical activity have always been known. The study of the relationship between exercise and the human phenotype that characterize aging and pathology appears imperative, given the beneficial effects that regular physical exercise exerts in promoting healthy aging and in alleviating age-related pathologies. Nevertheless, is important to elucidate the molecular bases at cellular and organismal level of this process. In humans and rodents, exercise has been shown to promote autophagic activity and lead to upregulation of basal autophagy levels in numerous tissues. From a general perspective, autophagy process is indispensable to improve exercise-induced metabolic level and potential health benefits of appropriate exercise. To this end, the multifunction of irisin in producing the beneficial effects of exercise at systemic level have become clear since its discovery in 2012. From this date, several studies suggested that circulating level of this myokine are significantly linked to physiological and/or pathological state. Recently, myokines have been mentioned as autophagy regulator, and starting in 2018, irisin has been significantly associated to autophagy induction in diverse cell types (i.e., myofibers, cardiomyocytes, hepatocytes, and pancreatic β-cell) (Figure 4). As we reviewed, these new important evidences contributed to clarify the beneficial role of exercise and irisin at cellular and molecular levels, at least on metabolic and heart related diseases. However, much are still needed to know about irisin and autophagy; this molecule remains appealing with a potential to bridge our knowledge gaps between exercise and the beneficial effects on aging and diseases. The rational next step is the study of the effect of irisin on the autophagic process in un-investigated cell types on which irisin exert its role, as well as adipocytes, neurons, bone cells, and immune cells. Nevertheless, it is necessary to identify the irisin receptor, which will shed light into its pathophysiology. It is also useful to keep in mind that the glycosylation state of irisin is crucial for its biological effect, whereas a recombinant nonglycosylated form of irisin has been used in the majority of the studies performed. As it was mentioned in Section 4, N-glycosylation of irisin is indeed fundamental to its secretion and the browning of adipocytes, and it seems plausible that N-glycans groups also affect the activities of irisin. Future studies should be aimed at determining other glycosylation-dependent irisin functions and potential glycosylation changes that occur in pathological conditions, associating the results to autophagy in different cell types or tissues, in order to open a new window to related novel properties and pharmacology for irisin.

## Figures and Tables

**Figure 1 ijms-21-07587-f001:**
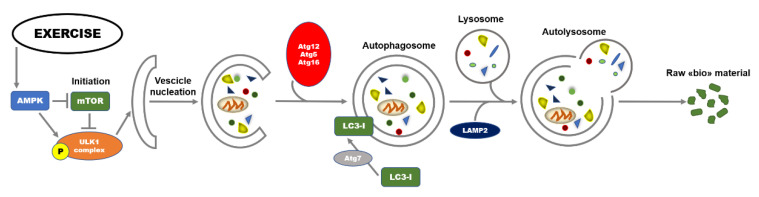
Overview of basic molecular mechanisms involved in initiation of exercise-induced autophagy. Autophagy is a lysosomal-mediated degradation process for primarily degrading damaged cells and dysfunctional organelles to recycle damaged or dysfunctional cellular contents, thereby providing the energy for nascent cells. Exercise, as a newly defined stimulus, can induce autophagy in cells. mTOR and AMPK are currently recognized as the sensors for detecting nutrition and energy efficiency in skeletal muscle, especially in exercise responses. mTOR and AMPK in the initial steps of the autophagy process through phosphorylation interaction with the ULK1 complex, respectively, whereas AMPK inhibits mTOR activity. AMPK: AMP activated protein kinase; Atg: autophagy related gene; LC3: microtubule-associated protein 1A/1B-light chain 3; mTOR: mammalian target of rapamycin; P: Phosphate; ULK1: Unc-51 like autophagy activating kinase 1.

**Figure 2 ijms-21-07587-f002:**
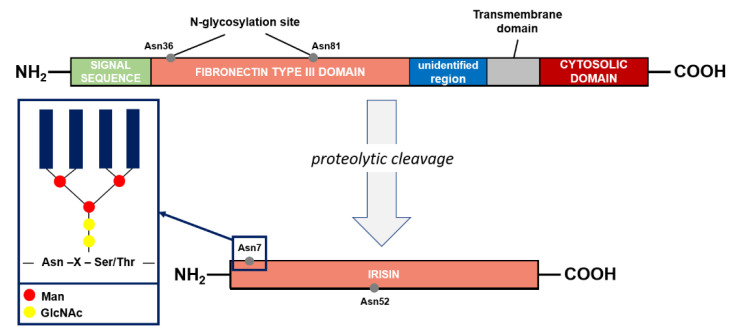
Fibronectin type III domain-containing protein 5 (FNDC5) structure and formation of irisin. The potential N-glycosylation sites are marked as gray dots. Asn, asparagine; GlcNAc, N-acetylglucosamine; Man, mannose; Ser, serine; Thr, threonine; X, any amino acid except proline.

**Figure 3 ijms-21-07587-f003:**
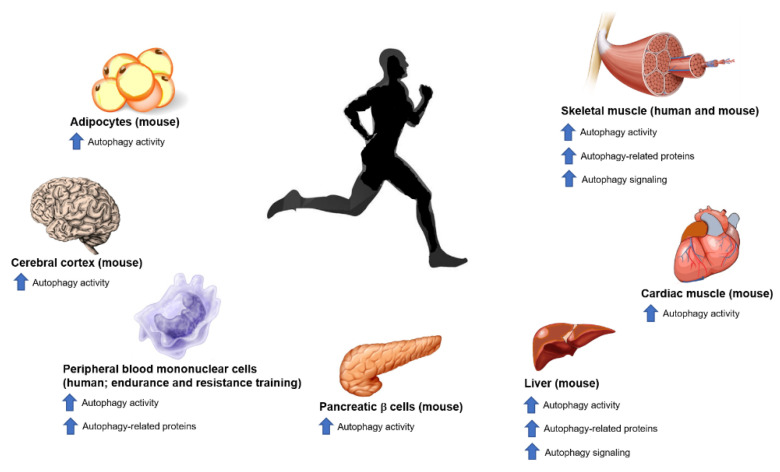
Effect of exercise on autophagy in multiple tissues.

**Figure 4 ijms-21-07587-f004:**
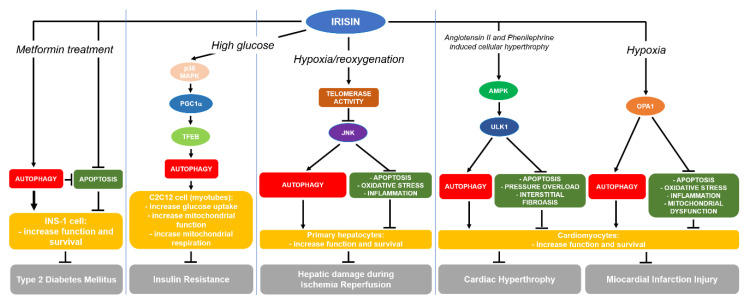
Effect of irisin on autophagy in multiple cells. AMPK: AMP activated protein kinase; INS-1 cell: insulinoma-1 cell; OPA1: dynamin-like GTPase optic atrophy 1; PGC1 peroxisome proliferator-activated receptor gamma coactivator 1-α; p38 MAPK: p38 mitogen-activated protein kinase; TFEB: transcription factor EB; JNK: c-Jun N-terminal kinase; ULK1: Unc-51 like autophagy activating kinase 1.

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
