# Peer review of "Irisin and Autophagy: First Update"

_ijms, 2020, doi:10.3390/ijms21207587_

Round 1

Reviewer 1 Report

Authors should discuss more in depth irisin in obesity and in metabolic disorders, as some controversy data exist. Irisin has been described to be increased in obesity in some studies and to correlate with insulin resistance (HOMA index) in metabolic syndrome, its role as biomarker could be also included. Authors should correct some typos. 

the text in fig4 is hard to read  

Author Response

Authors should discuss more in depth irisin in obesity and in metabolic disorders, as some controversy data exist. Irisin has been described to be increased in obesity in some studies and to correlate with insulin resistance (HOMA index) in metabolic syndrome, its role as biomarker could be also included.

Thank you for the comment. We have discuss more in depth the role of irisin in obesity, metabolic disorders considering also insulin resistance (HOMA index). Please see the text from line 292 to 311. 

Authors should correct some typos. 

Thank you. We have checked the manuscript for typos.

The text in fig4 is hard to read

Thanking you, we have improved the readability of the figure 4.   

Reviewer 2 Report

General comments:

  • The manuscript is interesting and well written. The only thing I could suggest is to explanation:
  • Why? What for autophagy exists in our organs. After reading this paper I am not convinced what for? I know that such process is present, I know its mechanism but don’t realize why ? and what for it is?
  • I strongly suggest Authors to improved and highlight this problems in paragraph about autophagy.
  • The same problem is with the exercise and autophagy.
  • We don’t know what is the main goal of its process and what mechanisms initiate the autophagy during exercise in SkM and different tissues. It plays the positive or negative role during exercise?
  • Please also explain what is the cause of autophagy and what is this process for.

Detailed comments:

  • Line 151 and in whole paper – I would suggest using hypokinesia instead disuse.
  • Line 268 – please also cite paper: Int J Environ Res Public Health. 2020;17(10):3589 doi:10.3390/ijerph17103589

Author Response

The manuscript is interesting and well written. The only thing I could suggest is to explanation:

Why? What for autophagy exists in our organs. After reading this paper I am not convinced what for? I know that such process is present, I know its mechanism but don’t realize why ? and what for it is? I strongly suggest Authors to improved and highlight this problems in paragraph about autophagy.

In accordance with the reviewer, we have improved the explanation on this matter from line 82 to 104 of the text.

The same problem is with the exercise and autophagy.

We don’t know what is the main goal of its process and what mechanisms initiate the autophagy during exercise in SkM and different tissues. It plays the positive or negative role during exercise? Please also explain what is the cause of autophagy and what is this process for.

We appreciate the comment. Please consider for the reply from line 341 to 361 of the text. Thank you.

Detailed comments:

Line 151 and in whole paper – I would suggest using hypokinesia instead disuse.

Thank you. We have substituted "disuse" with "hypokinesia".

Line 268 – please also cite paper: Int J Environ Res Public Health. 2020;17(10):3589 doi:10.3390/ijerph17103589

We have included. Thank you